# Polyunsaturated Fatty Acid-Enriched Lipid Fingerprint of Glioblastoma Proliferative Regions Is Differentially Regulated According to Glioblastoma Molecular Subtype

**DOI:** 10.3390/ijms23062949

**Published:** 2022-03-09

**Authors:** Albert Maimó-Barceló, Lucía Martín-Saiz, José A. Fernández, Karim Pérez-Romero, Santiago Garfias-Arjona, Mónica Lara-Almúnia, Javier Piérola-Lopetegui, Joan Bestard-Escalas, Gwendolyn Barceló-Coblijn

**Affiliations:** 1Institut d’Investigacio Sanitaria Illes Balears (IdISBa, Health Research Institute of the Balearic Islands), 07120 Palma, Spain; albert.maimo@ssib.es (A.M.-B.); karim.perez@ssib.es (K.P.-R.); javier.pierola@ssib.es (J.P.-L.); 2Research Unit, University Hospital Son Espases, 07120 Palma, Spain; 3Department of Physical Chemistry, Faculty of Science and Technology, University of the Basque Country (UPV/EHU), Barrio Sarriena s/n, 48940 Leioa, Spain; luciamartinsaiz9@gmail.com (L.M.-S.); josea.fernandez@ehu.es (J.A.F.); 4Quirónsalud Medical Center, 07300 Inca, Spain; dr.garfias@gmail.com; 5Son Verí Quirónsalud Hospital, Balearic Islands, 07609 Son Veri Nou, Spain; 6Hospital de Llevant, 07680 Porto Cristo, Spain; 7Department of Neurosurgery, Jimenez Diaz Foundation University Hospital, Reyes Catolicos Av., No 2, 28040 Madrid, Spain; mnclr23@gmail.com; 8Ruber International Hospital, Maso St., No 38, 28034 Madrid, Spain; 9Bioanalysis and Pharmacology of Bioactive Lipids Research Group, Louvain Drug Research Institute, Université Catholique de Louvain, 1200 Bruxelles, Belgium

**Keywords:** glioblastoma, MALDI-IMS lipidomics, temozolomide, modular gene expression, molecular subtypes, lipid metabolism

## Abstract

Glioblastoma (GBM) represents one of the deadliest tumors owing to a lack of effective treatments. The adverse outcomes are worsened by high rates of treatment discontinuation, caused by the severe side effects of temozolomide (TMZ), the reference treatment. Therefore, understanding TMZ’s effects on GBM and healthy brain tissue could reveal new approaches to address chemotherapy side effects. In this context, we have previously demonstrated the membrane lipidome is highly cell type-specific and very sensitive to pathophysiological states. However, little remains known as to how membrane lipids participate in GBM onset and progression. Hence, we employed an ex vivo model to assess the impact of TMZ treatment on healthy and GBM lipidome, which was established through imaging mass spectrometry techniques. This approach revealed that bioactive lipid metabolic hubs (phosphatidylinositol and phosphatidylethanolamine plasmalogen species) were altered in healthy brain tissue treated with TMZ. To better understand these changes, we interrogated RNA expression and DNA methylation datasets of the Cancer Genome Atlas database. The results enabled GBM subtypes and patient survival to be linked with the expression of enzymes accounting for the observed lipidome, thus proving that exploring the lipid changes could reveal promising therapeutic approaches for GBM, and ways to ameliorate TMZ side effects.

## 1. Introduction

Glioblastomas (GBM), the most aggressive type of astrocytoma, are the most frequent malignant primary brain tumor, accounting for 15% of all brain tumors and presenting a worldwide incidence of 3–4 per 100,000 people [1]. Standard treatment consists of surgery followed by chemotherapy and radiotherapy [2]. Unfortunately, this treatment renders a low median survival of fifteen months [3]. The reference chemotherapeutic agent is temozolomide (TMZ), a DNA alkylating agent. Regrettably, it induces severe side effects such as dizziness and blurred vision and, consequently, around 20% of patients treated with TMZ discontinue treatment [4]. Due to its high-rate of discontinuity and poor overall survival improvement, TMZ treatment is considered palliative (non-curative) [5].

As in many other cancers, the lack of models accurately mimicking genetic heterogeneity and tumor microenvironment hinders the study of glioblastoma [5]. However, the latest advances in genomics enabled TP53, ATRX, TERT, NF1, PTEN, and EGFR to be identified as GBM gene drivers, providing new insights into the development of GBM [6,7]. Moreover, mutations on IDH1 or IDH2 genes have been related to better patient outcomes [8]. Furthermore, three transcriptomic molecular subtypes are defined based on different genomic alterations: proneural, classical, and mesenchymal; thus enabling the identification of specific epigenetic alterations as well as molecular subtype-dependent interactions with the immune microenvironment [9]. In this sense, changes in cell-type composition, secreted extracellular vesicles, and soluble factors contribute to GBM microenvironment heterogeneity and are all involved in treatment resistance and tumor recurrence [10]. Interaction between glial cells and GBM cells is key to tumor growth and progression. While tumor-associated astrocytes can interact with endothelial cells and pericytes from the brain–blood barrier [11,12,13], they also appear to be involved in the limited response to radiation or temozolomide chemotherapy [13,14]. GBM cells also show the ability to communicate with innate immune cells (microglia), by changing their phenotype to enhance tumor growth and survival [5]. Hence, the definition of these molecular subtypes and the differential contribution of the microenvironment components is a critical step towards the development of more accurate treatment strategies [9].

Second to adipose tissue, the brain is the most lipid enriched organ in the body, particularly in membrane lipids, phospholipids, and sphingolipids. Cell membrane lipid composition, or membrane lipidome, includes hundreds of molecular species, each of which have specific roles that remain mostly unknown. Consistently, the lipidome has proven to be highly sensitive to pathophysiological processes, and alterations in cell lipid profile are associated with multiple pathophysiological processes such as differentiation [15,16,17,18], proliferation [19,20,21], and cancer development [22,23,24]. In fact, membrane lipid species are sensitive enough to be used as biomarkers for several cancer types, such as ovarian cancer, prostate cancer, and breast cancer [25,26,27]. Importantly, the irruption of imaging MS (IMS) techniques into the lipidomic field have clearly demonstrated how specific and sensitive the lipidome is to both physiological changes and pathological insults. In this context, brain tissue remains one of the most analyzed tissues by different IMS techniques, showing concise lipid species distribution between grey and white matter in human samples [28,29,30] and highlighting the potential of lipidomics to study GBM. Thus, imaging lipidomic techniques together with machine learning protocols are capable of rapidly classifying gliomas based solely on their lipid profiles, offering a potential tool for intraoperative examination and rapid classification [31].

In addition, the regulation of several genes involved in lipid metabolism, such as SCD and ELOVL6 which participate in mono- and polyunsaturated fatty acid (MUFA, PUFA, respectively) synthesis, is altered in GBM [32,33]. One of these PUFAs is arachidonic acid, the precursor of a large family of bioactive molecules intimately involved in inflammation. Importantly, a study analyzing human GBM identified significant correlations between the high expression of mPGES1 and PTGR1, enzymes involved in the synthesis of prostaglandins, and was related to poor patient survival [34]. Conversely, higher gene expressions of 15-HPGD, involved in prostaglandin catabolism, was associated with better outcomes in cancer patients, including GBM [34].

Taking into account the close relationship already established between lipid metabolism and GBM development, we considered the study of its lipidome and the impact on lipid composition of TMZ established with spatial resolution which could help to reveal new targets to treat GBM or address TMZ side effects.

## 2. Results

### 2.1. Impact of GBM and TMZ Treatment on Brain Tissue Lipidome

An ex vivo approach was employed to assess the impact of TMZ on GBM lipidome. Thus, surgically healthy and GBM biopsies were obtained from the same patient, immediately placed into DMEM-F12 cell culture medium and incubated in the presence or absence of TMZ (10 mg/mL, 4 h) or DMSO (vehicle). After the incubation period, biopsies were snap-frozen in the absence of cryoprotective substances, and healthy brain and GBM sections were analyzed using MALDI-IMS in negative- and positive-ion mode at 50 µm lateral resolution.

#### 2.1.1. Characterization of Healthy Brain and Glioblastoma Lipidome

First, the presence of the regions differing in their lipidome within the brain and GBM biopsies were investigated to establish the differences between healthy and GBM lipidome. We previously demonstrated the high correlation existing between IMS lipid clusters stablished by K-means and HD-RCA algorithms and histological tissue structures, cell types, or even cell pathophysiological states [18,22,35,36]. Briefly, the software considers the distribution of all lipids detected by applying a clustering or segmentation analysis. Then, it renders a visual representation where pixels with a similar or proximal lipid composition are grouped into the same region according to a clustering/segmentation algorithm. In this particular study, one of the challenges was to correlate lipid distribution with its anatomical counterpart. Unfortunately, the malignization process by itself already implies a loss of tissular architecture, while the fine architecture is inevitably compromised during incubation in a cell culture medium. Thus, we focused on distinguishing cells exhibiting a high proliferative rate, as highly proliferative cells are required for tissue maintenance and the progression of cancer [37,38]. Ki-67 (MKI67) is a cell proliferation marker and prognostic marker in GBM [39,40,41] involved in the perichromosomal layer during mitosis [42]. Therefore, MKI67+ staining was used to define regions of interest by immunofluorescence (IF). These MKI67+ regions were employed to identify the lipid cluster that overlapped the most with the most highly proliferative region in both healthy and GBM biopsies. While MKI67 staining revealed several regions with different IF intensity in tumor tissue, staining intensity was homogenous and low in healthy tissue (Figure 1).

Next, we compared the lipid profile of the highly proliferative areas of healthy and GBM samples. Lipidome analysis detected 124 different lipid species belonging to 11 lipid classes. Unsupervised PCA using all lipid species demonstrated that the lipidome of the highly proliferative regions (MKI67+ clusters) discriminates between healthy and GBM tissue (Figure 2a).

Statistical comparison of lipid classes demonstrated that phosphatidylethanolamine (PE), PI, and sphingomyelin (SM) were significantly increased in GBM (1.5-, 3.6-, 1.8-fold increase, respectively). Conversely, a statistical decrease was observed in sulfatide content in GBM tissue (2.9-fold decrease). It is worth mentioning that the values represented in Figure 2 account for the intensity detected during MALDI-IMS analysis, which depends on the ionization capability of each compound. Consequently, the intensity values of different lipid classes cannot be compared. For instance, sulfatides show high intensity values despite only accounting for approx. 4% of total lipids in white matter [44]. Thus, comparisons must be made exclusively within the same lipid classes.

Next, we analyzed the changes occurring at the molecular species level. Of special interest was the differential impact on diacyl PE and PE P- species. In the healthy brain, the most abundant PE diacyl species were 36:1, 40:6, and 38:4 (35.6, 13.9, and 11.3%, respectively). In GBM tissue, the most abundant species were 36:1, 38:4, and 36:2 (30.0, 13.4, and 10.1%, respectively). Total diacyl PE levels were greatly increased in GBM tissues (Figure 2), especially PE 34:0 and 36:4 (Figure 3a). In this study, we considered PE ether lipids were mostly PE plasmalogens as this is the most abundant subgroup in the brain [45]. While PE plasmalogen total levels were similar in both study groups, the disease had a profound effect at the PE plasmalogen molecular species level. In healthy tissue, PE P-36:2 and 38:4 followed by 34:1 and 40:6 species (20.8, 14.7, 13.7, and 10.2% of total PE plasmalogen, respectively) were the most abundant PE plasmalogen species. Conversely, 38:4, 36:4, 40:6, and 38:6 were the most abundant PE plasmalogen species in the MKI67+ GBM cluster (22.0, 14.8, 10.4, and 10.3% of total PE plasmalogen, respectively). Thus, the results showed a solid tendency for 36C- and 38C:PUFA-containing species to increase in detriment to 40C:PUFA- and MUFA/DUFA-containing species when comparing GBM to healthy tissue. This shift was statistically significant for 36:2 (20.8 vs. 3.6%), 36:4 (1.9 vs. 14.8%), 38:4 (14.7 vs. 22.0%), 38:6 (2.8 vs. 10.3%), and 40:5 (7.1 vs. 4.3%) (Figure 3b). Consistent with the literature, the most abundant PI species in healthy and GBM tissue was 38:4 (54.9 and 75.7%, respectively), which in turn was the only species to show a significant increase in GBM compared to healthy tissue (Figure 3c).

Regarding sphingolipids, the main SM species in the healthy brain were d36:1, d42:2, and d34:1 (41.0, 26.8, and 10.6%, respectively), whereas in GBM they were d34:1, d36:1 and d36.2 (37.5, 26.6, and 8.3%, respectively). The most striking changes between study groups were the sharp increase in d34:1 and d40:1 molecular species and the decrease in d42:2 in GBM tissue (Figure 3d). Finally, in the healthy brain, the main sulfatide species were d42:2, t42:1, and d42:1 (50.8, 15.9, and 11.3%, respectively), whereas in the tumor they were d42:2, d36:3, and d36:4 (23.9, 21.0, and 11.4%, respectively). In this study, sulfatide species showed the most profound changes in composition in GBM tissue. Compared to healthy tissue, GBM presented lower levels of d42:2, d42:1, d44:2, and t42:1, with significant values for the latter three, while d34C and d36C-species were increased, significantly for d36:2 (Figure 3e).

#### 2.1.2. Effects of Temozolomide Treatment on the Lipidome of the Proliferative Areas in GBM and Healthy Brain

Although TMZ is the standard care treatment for GBM, the impact it might have on the lipidome remains unknown. Thus, to further understand this aspect, we compared the lipid profiles of the clusters overlapping with the most proliferative areas, i.e., the MKI76+ areas in the biopsies incubated with TMZ.

PCA revealed that the two experimental groups, TMZ treated and non-treated GBM, could be successfully differentiated based on the lipid profile of the proliferative regions (Figure 4). However, TMZ treatment did not bear a statistically significant impact at the level of lipid class, and only PE 38:5 was significantly increased at the molecular species level (Figure 5).

#### 2.1.3. Temozolomide Exerts Multiple Effects over the Healthy Brain Lipidome

One of the most relevant clinical issues of GBM treatment is the discontinuity of chemotherapy due to side effects. In this study, we investigated the impact of TMZ on healthy brain lipid composition. The results showed that the treatment induced changes at the level of both lipid classes and molecular species composition. Thus, PCA clearly separated the study groups based on lipid class composition (Figure 6a). Statistical comparison revealed a significant increase in the treated group in PI (2.2-fold increase) and a decrease in hexosylceramide, and sulfatides (1.8- and 1.6-fold decrease, respectively).

Regarding diacyl PE species, the most abundant species in healthy TMZ treated brain were 36:1, 40:6, and 38:4 (27.1, 23.2, and 15.8%, respectively). In this class, the only significant changes were a slight increase in 34:0 (3.2 vs. 5.5%) and a decrease in 38:1 (6.1 vs. 4.0%) (Figure 7a). The most abundant PE P- species were 40:6, 38:4, and 40:4 (23.5, 20.4, and 11.5%, respectively). Treatment increased many of the PUFA-containing PE P- species (Figure 7b), significantly for 38:4, 38:6, and 40:7 (14.7 vs. 20.4%, 2.8 vs. 6.5%, and 1.8 vs. 3.2%, respectively). These increases in PUFA-containing species were compensated by a decrease in MUFA and DUFA-containing ones, which were significant for 34:1 and 36:2 (13.7 vs. 5.0%, and 20.8 vs. 7.3%, respectively). Finally, TMZ treatment raised the PUFA-containing PI levels significantly for 38:4 and 38:5 (54.9 vs. 80.4%, and 5.1 vs. 6.3%) (Figure 7c).

Finally, TMZ treatment did not drastically affect SM lipid composition in healthy brain. The main SM species in treated tissue were d36:1, d38:1, and d42:2 (50.1, 12.4, and 12.1%, respectively). TMZ decreased the level of d42:2, d42:1, and d44:2 compared to vehicle-treated tissue (Figure 7d). Meanwhile, sulfatides remained almost unchanged, with only d36:2 sulfatide increasing significantly in the TMZ group (Figure 7e).

In summary, our results demonstrated changes in various lipid species such as phosphatidylinositols (PI), sulfatides, and phosphatidylethanolamine plasmalogens (PE P-) in GBM tissue. Thus, PE P- species were the most affected by the tumor transformation, showing a general increase in PUFA-containing species. On the other hand, TMZ treatment of healthy and tumor brain tissue revealed that, in short-term treatments, this compound barely affected the GBM lipidome. Conversely, this compound induced multiple changes in the healthy brain, with PE plasmalogen species the most affected.

### 2.2. Impact of GBM on Lipid Enzymes at the Transcriptional Level

The brain represents one of the tissues showing the most singular lipid profile and, as expected, such a complex process as tumorigenesis bears a profound impact on it. In order to outline the mechanisms accounting for the altered lipid phenotypes observed in GBM and to delve into the regulatory mechanisms, we interrogated a publicly accessible transcriptome database, specifically, the TCGA-GBM transcriptome database. Co-expression modular analysis (CEMiTool) [46] was performed by applying the molecular subtype labels described by Verhaak et al. [47] on the TCGA-GBM AffyU133a dataset (Figure 8).

The CEMiTool analysis returned seven modules wherein it was possible to distinguish several hub genes (Appendix A). Using the normalized enrichment score (NES), we identified six of the modules correlating positively with some of the four molecular subtypes: Classical, Mesenchymal, Proneural, and Neural (Figure 8, Appendix A). The M1 module showed a high correlation with the Classical molecular subtype module, the M2 and M4 modules correlated with the Mesenchymal, the M5 and M6 modules aligned with the Proneural, while the M3 module did so with the Neural subtype and Normal samples. Detailed information regarding the complete gene composition and gene ontology (GO) analysis of each molecular subtype correlated module can be found in Appendix A, respectively.

Based on the lipidomic differences observed between Healthy vs. GBM study groups (Figure 2), we decided to interrogate the genes coding for enzymes involved in PUFA- and sphingolipid metabolism. Among them, genes coding for FABPs, ELOVL, inositol polyphosphate-5-phosphatase F (INPP5F), secreted PLA2s, PTGS2 (COX2), ALOXs, diacylglycerol kinase B (DGKB), and UDP-galactose-ceramide galactosyltransferase (UGT8) were identified (Table 1). These results would agree with a relevant contribution of the expression profile of these lipid genes in each of the GMB molecular subtype phenotypes. An overview of the function of these enzymes is included in Appendix A.

The Classical and Mesenchymal subtype modules contained several enzymes involved in fatty acid uptake and intracellular signaling (FABPs), fatty acid release from phospholipids by hydrolysis (PLA2s), PUFA synthesis (ELOVL2), and eicosanoid synthesis (PTGS2, ALOXs), which would be in line with the increase in PUFA-containing PE P- and PI species. On the other hand, the Neural subtype was enriched with INPP5F and DGKB, both involved in phospholipid metabolism homeostasis and intracellular signaling by hydrolyzing one phosphate from the phosphoinositides and adding one phosphate to DG, respectively. Finally, the module enriched in the Proneural subtype contained UGT8, a sphingolipid related enzyme necessary for the synthesis of the myelin sheath. Hence, we analyzed the patient survival data associated with the expression levels of the genes included in Table 1, to further understand the implication of these enzymes in GBM progression (Figure 9).

Importantly, we discovered that the expression of five of the genes, enriched in Classical and Mesenchymal subtypes, were significantly associated with poor overall patient survival (PLA2G5, FABP7, ELOVL2, PLA2G2A, and ALOX5AP) (Figure 9). These genes are tightly involved in fatty acid metabolism, phospholipid synthesis and remodeling, and eicosanoid synthesis according to GO biological process enrichment (Appendix A). Moreover, the genes in Figure 9 were also statistically significant in the disease-specific survival tests, as well as ALOX15B (M2-Mesenchymal) and UGT8 (M6-Proneural) (Appendix A). Meanwhile, the rest of the genes included in Table 1—FABP5, PTGS2, INPP5F, and DGKB—did not show any statistical association with survival outcome.

Finally, it is well established that DNA methylation is an additional mechanism to regulate gene expression. Hence, we used the TCGA-GBM public database to assess the methylation status of the lipid genes that were significantly associated with worse overall survival (Figure 9). This interrogation revealed a clear relationship between methylation status and expression levels in three genes in particular: *PLA2G5*, *FABP7*, and *ALOX5AP* (Figure 10). PLA2G5 expression was lower in the Proneural subtype compared to the rest of the subtypes (Figure 10a). The cg2433549 methylation probe for PLA2G5 showed higher methylation levels in the Proneural subtype and lower levels in the other subtypes, especially in the Classical (Figure 10b). Similarly, FABP7 displayed lower gene expression levels and higher cg18555555 methylation levels in the Proneural subtype, while the Classical exhibited high gene expression and lower methylation levels compared to the rest (Figure 10a,b). ALOX5AP showed a lower expression level in the Proneural and Classical subtypes and higher cg08529529 methylation levels (Figure 10a,b). Conversely, the Mesenchymal subtype revealed higher gene expression of ALOX5AP and lower methylation level of cg08529529 compared to the rest of subtypes (Figure 10a,b).

To summarize, a set of lipid-related enzymes was identified in specific co-expression modules enriched in different molecular subtypes. Some of these lipid-related enzymes showed statistical association with poor overall and disease-specific survival. Moreover, PLA2G5, FABP7, and ALOX5AP subtype-dependent gene expression agreed with their methylation status, i.e., less expressed in the genes with higher methylation levels.

## 3. Discussion

The data generated herein by MALDI-IMS analysis demonstrates that the areas of high proliferation within the GBM tissue can be identified based solely on the specific lipidome, and that these areas possess an altered lipid fingerprint compared to the healthy brain. Further, the results show that the brain lipidome is sensitive enough to undergo multiple changes when healthy tissue is subjected to TMZ treatment. Interestingly, current knowledge of the mechanism of action of this drug implies the preferable induction of mutations in tumor tissue through the methylation of adenines and guanines [49]. To our knowledge, this is the first study providing insights into the side effects of chemotherapy treatments on the brain lipidome, revealing an undescribed effect of TMZ on neural tissue. While more studies would be needed to understand the underlying mechanisms accounting for our observations, the short-time treatment used herein would exclude the possibility of the drastic changes observed in healthy brain being the consequence of the impact of TMZ on gene expression.

Several studies found in the literature demonstrate the potential of lipidomic analysis in the study of GBM. Thus, gliomas can be rapidly classified based on their lipid profiles established by DESI-IMS, offering a potential tool for intraoperative examination and rapid classification of surgical specimens [31]. In addition, differences in glycosphingolipid and triglyceride have been established in ectopic and orthotopic human xenografts models, sustaining the critical role of some lipids in tumor growth [50]. Furthermore, ELOVL2, an important enzyme in PUFA synthesis, is involved in maintaining EGFR signaling and GBM proliferation, through its contribution to membrane composition [51]. Moreover, LXRβ up-regulates cholesterol biosynthesis in patient-derived neurospheres, enabling glioma cells to proliferate and survive at high cell densities even when cholesterol is high [52,53]. Altogether, these recent studies reinforce the key role of lipid metabolism in GBM progression.

Our results demonstrate that each phospholipid class responded differently in GBM tissues, with PE plasmalogens changing the most. Interestingly, PE diacyl species barely changed, reinforcing the evidence that the presence of vinyl-ether linkage entails different biological roles and regulatory mechanisms compared to diacyl phospholipids [22,54]. Compared to PE-diacyl, PE-vinyl ether species are highly enriched in PUFAs, mainly in arachidonic and docosahexaenoic acid (20:4n-6 and 22:6n-3) [22,54], which are found almost exclusively at the sn-2 position [55]. In general, the tumors analyzed displayed a clear increase in PUFA-containing PE plasmalogens, which is consistent with previous studies [18,56]. Therefore, the increase in PUFA-containing species but not in total PE P- described in the GBM samples would be in line with these premises.

The relationship between arachidonic acid and cell viability is complex. Free arachidonic acid induces neuronal toxicity in isolated healthy neurons [57]; in glial cell lines however, it promotes survival [58]. On the other hand, this fatty acid has antiproliferative capacities for several cancer types [59], including glioblastoma [60], while also serving as a substrate for important bioactive lipids. Our previous studies in colon cancer, demonstrated differential PE plasmalogen composition in cancer tissue accompanied by the disrupted expression of their rate-limiting proteins (i.e., FAR1, FAR2, GNPAT, and AGPS) [22,61], which could be explained by the role of plasmalogens as hubs of bioactive molecules. Thus, tumors would enhance the synthesis of plasmalogens to feed their need for bioactive lipids [62,63]. Consistently, cPLA2α, the main enzyme releasing arachidonic acid from phospholipids, is upregulated in GBM and TMZ-resistant GBM cells [64]. Moreover, the interrogation of transcriptomic databases showed that the altered expression of several phospholipases A2 is a common feature in different molecular subtypes of GBM (Table 1, Figure 10). Finally, lower expression of prostaglandin synthesis enzymes, particularly PGE2S, and a lower concentration of PGE2 and PGF2α are related to better patient outcomes and lower tumor grade [34].

The main PI species, PI 38:4, represents another bioactive precursor hub that significantly increased in GBM tissue compared to healthy tissue. The main species contributing to PI 38:4 by far is PI 18:0/20:4, consequently becoming a source of arachidonic acid-derived molecules. Furthermore, PI species are the precursors of phosphoinositides, members of a canonical family of signaling molecules linked through a highly regulated cycle of de-phosphorylation reactions. In fact, the PI3K-AKT signaling pathway is upregulated in GBM [65] and is necessary for the development of GBM in mouse model tissue [66]. Despite a certain level of specificity of PI kinases towards arachidonic acid-containing PI [67], knowledge regarding PI kinase and PI hydrolyzing enzyme specificity towards specific lipid species remains still very limited. Therefore, the increase in PI 38:4 observed in GBM tissue may be the result of the tumor attempting to meet the needs of the tumor tissue once phosphoinositide signaling is triggered.

Sulfatide composition, a phospholipid class particularly abundant in the brain, was also altered at the level of lipid species in GBM. Our results revealed lower levels of sulfatides in tumor tissue compared to healthy tissue. This outcome agrees with previous studies showing lower amounts of sulfatides in grade IV astrocytomas than low-grade astrocytomas [31]. Sulfatide species with C18 fatty acid chain length tended to increase in tumor tissue to the detriment of fatty acids with longer chains. Interestingly, neurons and astrocytes are enriched in sulfatides containing shorter fatty acids compared to oligodendrocytes (C18 vs. C22/C24) [68]. This observation can be partially explained by a greater presence of white matter in healthy tissue, enriched in neurons and axons than in tumor tissue, in which most cells would originate from the glia. Despite the lack of major changes in SM and sulfatide species after TMZ treatment, other studies demonstrated an increase in short-chain fatty acid sphingolipids in glioma cells transfected with p53 and treated with a chemotherapeutic agent (SN-38) [69]. The shorter treatment times herein and the use of a different chemotherapy agent could explain this divergence in results.

All in all, there is no doubt that addressing the reasons underlying the global increase in PUFA levels could help to understand the side effects induced by TMZ and to counteract them. Previous reports already linked changes in the expression of lipid-related enzymes with GBM onset. Thus, SCD1, a key enzyme in the synthesis of unsaturated fatty acids, is essential for the survival of GBM cells [32], while ELOVL6, involved in the elongation of fatty acids, is overexpressed in GBM patient tissue [33]. Furthermore, the expression of 15-HPGD, coding for the enzyme in charge of degrading bioactive lipids, is related to longer survival of GBM patients [34].

The application of -omics techniques provides resourceful information to classify patients according to their molecular signature [70]. Hence, genomic, epigenomic, and transcriptomic analysis allows a precise stratification of GBM samples [71]. For example, the Proneural molecular subtype, which presents better survival data, frequently bears a mutation in the IDH (isocitrate dehydrogenase) gene, a well-established prognostic marker. Moreover, IDH-mutants commonly manifest the glioma CpG island methylator phenotype (G-CIMP), which is also associated with a survival advantage [72,73]. A more detailed analysis showed that in fact, enhanced survival is determined by G-CIMP level [74]. On the other hand, patients with higher MnSOD (SOD2) protein expression are most likely to be IDH1 wild type, with poor overall survival and early progression-free survival [75]. In our co-expression analysis, SOD2 was found to be one of the genes defining the M2-mesenchymal subtype (Appendix A), which is characterized by poor survival, extensive necrosis, inflammation, angiogenesis, highly cell-enriched tumor micro-environment, and resistance to different therapies [76]. Further, the TCGA-GBM database shows that low SOD2 (MnSOD) expression is associated with mutant IDH1 (R132H), proneural subtype, and more specifically with G-CIMP status. Interestingly, a recent observational prospective study describes how GBM patients with wild-type IDH1/2, with a Karnofsky Performance Score >80, treated with concomitant radio-chemotherapy and subsequent chemotherapy with TMZ—which presented non-local recurrence—have poorer overall survival than patients with local recurrence [77]. It might be of great interest to measure the expression of mesenchymal markers [78] in the subset of non-local recurrence patients, in order to describe a positive correlation. With this in mind, the genomic and metagenomic analysis of lipid-related enzymes could help in understanding how the observed lipid changes are regulated at the gene level.

In this study, we employed the molecular subtype classification established by Verhaak et al. [47] to delve into the co-expressed genes coding for lipid enzymes that could account for the GBM lipidomic phenotype described. The analysis using TCGA-GBM databases identified nine lipid-related genes whose expression turned out to be regulated in a molecular subtype-dependent manner. Further, five of these, namely PLA2G5, PLA2G2A, FAPB5, FABP7, ELOVL2, and ALOX5AP, were significantly associated with poor overall and disease-specific survival. PLA2G5, FABP7, and ALOX5AP gene expression levels also showed conspicuous epigenetic regulation according to the specific gene region methylation levels. Particularly remarkable was the high methylation of FABP7 and PLA2G5 in the Proneural subtype, together with lower gene expression. Conversely, FABP7 and PLA2G5 exhibited low methylation and higher gene expression in the Classical subtype when compared to the other subtypes. Finally, the ALOX5AP gene showed lower methylation and higher gene expression in the Mesenchymal subtype compared to the others.

The Classical subtype gene signature is characterized, among others, by elevated EGFR and NOTCH3 expression. Recent studies have strengthened the involvement of ELOVL2 in EGFR signaling maintenance and GBM proliferation, through its contribution to PUFA synthesis and membrane composition. Further, ELOVL2 expression was also found to be associated with poor survival [51]. FABP7, a protein involved in the mobilization and transport of fatty acids and with a high affinity for arachidonic and docosahexaenoic acid, is involved in brain development and has been described to improve cell migration and infiltration in malignant glioma cells [79,80]. In addition, FAB7 modulates the activity of PKC under arachidonic and docosahexaenoic acid supplementation in GBM patient-derived neurospheres and cell lines [81]. These studies support the role of FABP7 in the increase in PUFA-containing phospholipids and their engagement in GBM progression, especially in the Classical subtype. Ultimately, the activity of PLA2G5, a secreted PLA2 with a high affinity for unsaturated fatty acids, may also be involved in the PUFA metabolism. Meanwhile, the elevated methylation of ALOX5AP in the Proneural subtype and its lower methylation in the Mesenchymal suggest the involvement of the eicosanoid metabolism in stromal cells, which could account for tumor microenvironment enrichment in arachidonic acid. Proneural to Mesenchymal transition is a described mechanism for resistance to chemotherapy in GBM relapse. Thus, radiotherapeutic treatment favors the presence of a Mesenchymal from a Proneural phenotype. The transformation of Proneural or selection of Mesenchymal cells, more resistant to radiotherapy, could represent the mechanisms responsible for this acquired resistance [82,83]. In this scenario, the pivotal expression and methylation levels of ALOX5AP strongly indicate a relevant role of arachidonic acid-containing phospholipids and the derived metabolism in the tumor microenvironment and GBM progression.

## 4. Materials and Methods

Materials and reagents: 2-Mercaptobenzothiazole (MBT) and 1,5-diaminonaphtalene (DAN), hematoxylin and eosin for histological staining, Ki67 primary antibody marked with FITC (ThermoFisher Scientific, Waltham, MA, USA), DMEM-F12, FBS, penicillin–streptomycin, and TMZ were purchased from Sigma–Aldrich (Barcelona, Spain).

Human sample collection: Sample collection for this study was specifically approved by the Ethics Research Committee of the Balearic Islands (nº IB 3626/18 PI). Informed consent in writing was obtained for each patient enrolled in the study. Four patients harboring brain tumors suggestive of GBM, newly diagnosed after neurological symptoms in which surgical resection or open biopsy were indicated, were included. Anatomopathological analysis confirmed the diagnosis of glioblastoma in the four patients recruited. Patients received the pharmacy-compounded solution of 5- aminolaevulinic acid 1 h before anesthetic induction. Total dose was according to patient weight (20 mg/kg). The Pentero^®^ (Zeiss^®^, Oberkochen, Germany) surgical microscope with BLUE 400TM integrated fluorescence module was used for surgical resections. Between 3 and 4 h after administration of 5-aminolevulinic acid, positive tumor samples and healthy samples were obtained during surgical resection from the surgical margin and immediately incubated in DMEM-F12 in the presence or absence of TMZ (10 mg/mL, Sigma–Aldrich (Barcelona, Spain)) for 4 h at 37 °C and 5% O_2_. These were then collected and stored at −80 °C until processing.

Histological sections: Sections 10 μm thick were obtained with a cryostat (Leica CM3050S, Leica Biosystems, Wetzlar, Germany) at −20 °C without using cryoprotective substances or embedding material. Sections were placed on plain glass microscope slides for MALDI-IMS analysis and consecutive sections for IF analysis were placed on positive charged adherent plain glass microscope slides. Samples were stored at −80 °C until subsequent MALDI-IMS or immunofluorescence analysis.

Immunofluorescence analysis: Histological sections were fixed with −20 °C prechilled 100 µL methanol–acetone (50/50, *v*/*v*) and then incubated with 1:500 MKI67 primary antibody marked with FITC (10 mg/mL) (ThermoFisher Scientific, Waltham, MA, USA) in 0.2% BSA-PBS, following the previously described protocol in Bestard-Escalas et al. [18]. Nuclei staining was performed with 4′6-diamidino-2-phenylindole (DAPI, 1 mg/mL, BD biosciences, Barcelona, Spain) at 1:10,000 dilution for 1 min at room temperature. Finally, samples were observed with Axioscope Cell Observer microscope and/or Zeiss LSM 700 confocal microscope (Carl Zeiss, Oberkochen, Germany).

MALDI-IMS analysis: A total of 32 (four sections for each treatment and ionization mode) histological sections obtained from four different patients were prepared and analyzed by MALDI-IMS as described in Garate et al. [36]. Briefly, MBT or DAN were used as matrix for positive- or negative-ion detection, respectively, and deposited with the aid of our in-house designed sublimator device, which allows perfect control of all the parameters involved in the sublimation process [84]. Sections of brain biopsies from different individuals were scanned in positive- and negative-ion mode using the orbitrap analyzer of a MALDI-LTQ-Orbitrap XL (Thermo Fisher, San Jose, CA, USA). The MALDI source used in this study was the one provided by the manufacturer, which is equipped with a N2 laser (LTB, Berlin, model MNL 100, 100 μJ max power, elliptical spot, 60 Hz repetition rate), and a very simple optical arrangement, consisting of two mirrors and a single focusing lens of f = 125 mm.

Data were acquired with a mass resolution of 60,000 in the scanning range of 550–1000 for negative-ion mode and 480–100 Da for positive-ion mode. Two microscans of 10 laser shots were recorded for each pixel and the raster size used was 50 microns. Spectra were aligned and analyzed using in-house programs developed in Matlab (MathWorks, Natick, MA, USA). Lipid assignment was based on the comparison between the experimental *m*/*z* and the species in the software’s database (<33,000 lipid species plus adducts) and in the lipid maps database (www.lipidmaps.org (last accessed on 10 January 2022)). Mass accuracy always measured better than 9 ppm and was typically better than 3 ppm. In this type of analyzers, mass accuracy depends somehow on the intensity of the peaks, therefore, the *m*/*z* with higher intensity present better mass accuracy. For *m*/*z* channels with several lipid assignments, “On-tissue” MS/MS and MS3 was carried out in order to unequivocally assign them.

For the sake of clarity, only species present in at least 80% of the analyzed samples were considered for further statistical analysis.

Interrogation of GBM gene expression and methylation datasets: Human TCGA GBM Affymetrix U133a and Methylation27k datasets were interrogated using Xena Browser. The data relative to the selected genes shown in Figure 10 were downloaded and statistical differences were measured independently.

Co-Expression Modules identification Tool (CemiTool): CEMiTool was used to identify co-expression of lipid related genes associated with the GBM molecular subtypes, according to Verhaak et al. [47]. To this end, the TCGA-GBM AffyU133a dataset and associated clinical data were downloaded and analyzed. The CEMiTool analysis returned seven modules with a different gene number and composition. Six of the modules were positively correlated, according to the normalized enrichment score (NES), with some of the four molecular subtypes applied as phenotype labels (Figure 8, Appendix A). The parameters used in CEMiTool were the following: value of Beta chosen = 9; Pearson correlation coefficient, dissimilarity threshold used as cutoff on hierarchical clustering = 0.8; similar modules were merged, the number of module returned = 8; area under curve / total area in the Beta vs R squared graph = 0.802; determination coefficient (“scale-freeness” of the resulting network) = 0.939.

Gene ontology (GO) analysis: This was carried out using the DAVID bioinformatic resources 6.8 server [85].

Statistical analysis: To establish tissue clusters, an in-house programmed clustering algorithm was used, setting the number of segments from 2 to 8. To statistically evaluate the differences in the lipid fingerprints between the identified areas, *t*-test, ANOVA, and Post Hoc analysis were computed using SPSS Statistics 25.0 for Windows (IBM, Armonk, NY, USA). PCA analysis and separation models were carried out by Orange Biolab 2.7.8 (Ljubljana, Slovenia) [86]. For TCGA GBM gene expression and methylation, multiple comparison ordinary one-way ANOVA with post-hoc Tukey test was computed using GraphPad Prim (version 8.0).

## 5. Conclusions

Altogether, the present study provides solid evidence regarding the sensitivity of the membrane lipidome in the development of GBM and describes the multiple effects occurring at the level of lipid composition in brain tissue upon TMZ treatment. The results suggest a scenario where, over short time periods, the tumor tissue lipidome is initially impervious to treatment, while the healthy brain is more sensitive to it. The unexpected sensitivity of healthy tissue to this treatment could be related to the side effects of blurry vision or dizziness that eventually lead to high discontinuity rates. There is no doubt that knowledge of the side effects could help reduce discontinuity rates, which currently affect 20% of the patients, and increase the effectiveness of future therapies. However, this study contains some limitations; namely, the number of samples included was limited, due to the difficulty in obtaining these types of samples and the challenges associated with the ex vivo model, which can rapidly compromise tissue structure. It is in the light of these limitations, that the results of the in silico approach gain more relevance, as the transcriptomic and genomic results support the lipidomic results described herein. Thus, the interrogation of the TCGA-GBM transcriptome database highlighted the role of PUFAs in GBM progression, especially in the Classical and Mesenchymal subtypes. The description of how lipid enzymes are differently regulated according to molecular subtype was discovered to be in line with the changes in PUFA-containing phospholipids described in this work. This genomic analysis revealed a specific lipid gene signature depending on the molecular classification of GBM. Globally, the results showed newly coordinated lipid–genetic changes that could set the base for other approaches to GBM treatment.

## Figures and Tables

**Figure 1 ijms-23-02949-f001:**
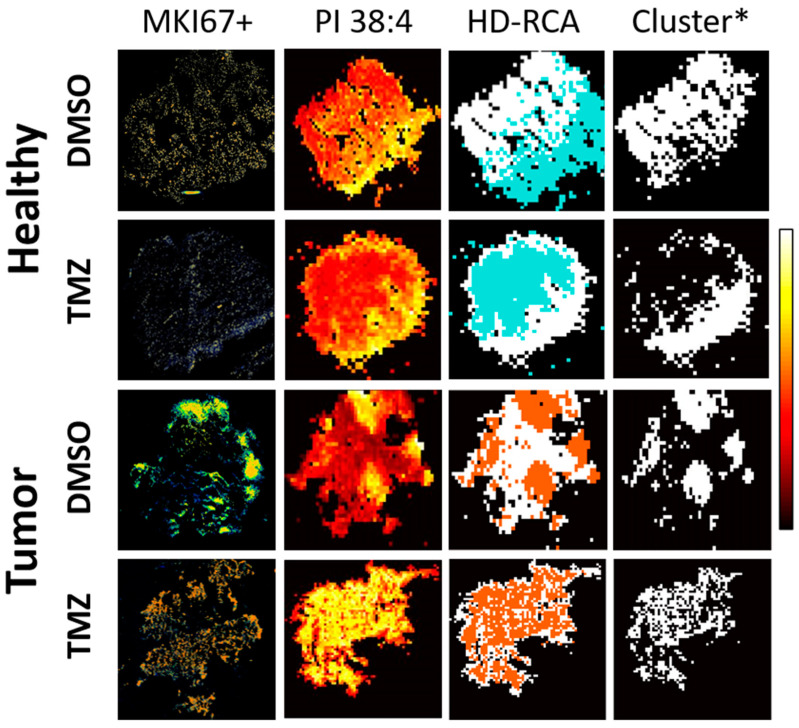
Brain proliferation zones and MALDI-IMS clustering comparison. Representative images of IF and MALDI-IMS for healthy and tumor treated and non-treated are shown. Healthy brain and GBM biopsy sections were prepared after incubating fresh biopsies in DMSO (vehicle) or TMZ (10 mg/mL, 4 h) and analyzed by MALDI-IMS at 50–100 µm lateral resolution. Brain proliferation zones were determined by MKI67+ IF staining and used to select the MALDI-IMS cluster, generated by HD-RCA from consecutive tissue sections [43]. HD-RCA clustering enabled the identification of the IMS regions of interest (Cluster*) with greater correlation (evaluated by direct visual inspection) with the MKI67+ IFs, based on the similarity of lipidomic content in each MALDI-IMS experiment. Proliferation zones are marked in orange in MKI67+ IFs. DAPI was used as a nucleus marker (marked in blue on IF MKI67 images). The distribution of PI 38:4 (885.55 *m*/*z*) is shown as a representative MALDI-IMS lipid distribution. Color scale indicates the intensity of the PI 38:4 -H distribution (0, black; 1, white). HD-RCA number of segments was set from 2 to 5, with prior background noise filtration using in-house MATLAB algorithms [43].

**Figure 2 ijms-23-02949-f002:**
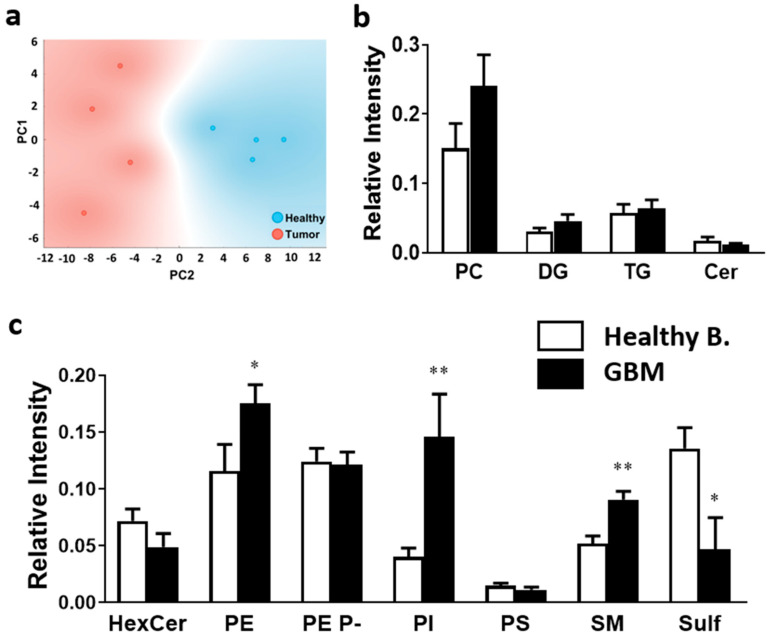
Impact of tumorigenesis on MKI67+ clusters at the level of lipid class composition. Healthy brain and GBM biopsy sections were prepared after incubating fresh biopsies in DMSO (vehicle) or TMZ (10 mg/mL, 4 h) and analyzed by MALDI-IMS at 50 µm lateral resolution. (**a**) PCA analysis showed a clear separation between healthy brain (blue), and GBM (red) tissue, implying distinct lipid fingerprints. Variance is explained by the two components: 80.4%. (**b**) Relative intensity variation of the main membrane lipid class was analyzed in the positive-ion mode in healthy brain and GBM. (**c**) Relative intensity variation of the main membrane lipid class was analyzed in the negative-ion mode in healthy brain and GBM. Values are expressed as relative peak intensity normalized to total ion current and represent the mean ± SEM (*n* = 5). Statistical analysis was assessed using *t*-test analysis. * *p* < 0.05; ** *p* < 0.01. Abbreviations: HexCer, hexosylceramides; PE, phosphatidylethanolamine; PE P-, PE plasmalogen; PG, phosphatidylglycerol; PI, phosphatidylinositol; PS, phosphatidylserine; SM, sphingomyelin; Sulf, sulfatide; PC, phosphatidylcholine; DG, diacylglycerol; TG, triacylglycerol; Cer, ceramide.

**Figure 3 ijms-23-02949-f003:**
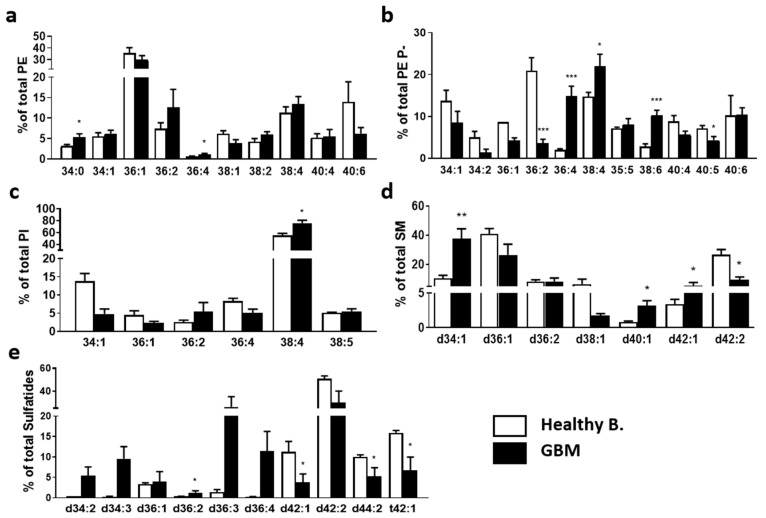
Impact of tumorigenesis on MKI67+ clusters at the level of molecular species composition. Healthy brain and GBM biopsy sections were prepared after incubating fresh biopsies in DMSO (vehicle) or TMZ (10 mg/mL, 4 h) and analyzed by MALDI-IMS at 50 µm lateral resolution. Each graph represents the percentage of each molecular species within each phospholipid class. (**a**) PE, (**b**) PE P-, (**c**) PI, (**d**) SM, and (**e**) Sulf. Values are expressed as the percentage of total fatty acid (mole%) and represent the mean ± SD, *n* = 5. For simplicity, species accounting for less than 5% were not included in the graph. Detailed results of all lipid species identified are included in Appendix A. Statistical significance was assessed using *t*-test analysis, * *p* < 0.05; ** *p* < 0.01; *** *p* < 0.001. Abbreviations: PE, phosphatidylethanolamine; PE P-, PE plasmalogen; PI, phosphatidylinositol; SM, sphingomyelin; Sulf, sulfatide.

**Figure 4 ijms-23-02949-f004:**
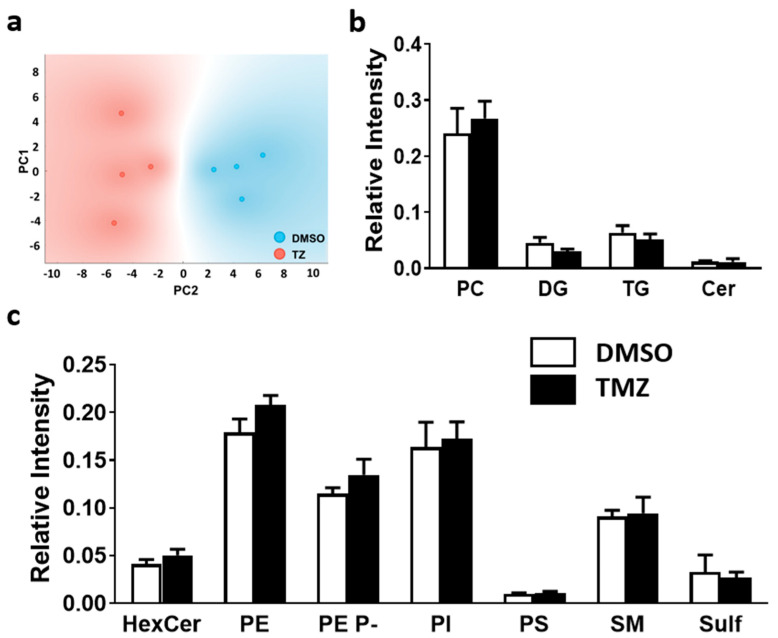
Impact of TMZ treatment on membrane lipid class composition of the MKI76+ region in GBM. GBM biopsy sections were prepared after incubating fresh biopsies in DMSO (vehicle) or TMZ (10 mg/mL, 4 h) and analyzed by MALDI-IMS at 50 µm lateral resolution. (**a**) PCA analysis of the proliferation (MKI67+) clusters identified in non-treated GBM (blue), and TMZ-treated GBM (red). Variance is explained by the two components: 71.3%. (**b**) Lipid class relative intensity variation analyzed in the positive-ion mode of the proliferative regions in non-treated GBM (DMSO) and TMZ treated GBM. (**c**) Lipid class relative intensity variation analyzed in the negative-ion mode of the proliferative regions in non-treated GBM (DMSO) and TMZ treated GBM. Values are expressed as the relative abundance of peaks and represent mean ± SEM (*n* = 4). Statistical analysis was assessed using *t*-test analysis. Abbreviations: HexCer, hexosylceramides; PC, phosphatidylcholine; PE, phosphatidylethanolamine; PE P-, PE plasmalogen; PI, phosphatidylinositol; PS, phosphatidylserine; SM, sphingomyelin; Sulf, sulfatide; PC, phosphatidylcholine; DG, diacylglycerol; TG, triacylglycerol; Cer, ceramide.

**Figure 5 ijms-23-02949-f005:**
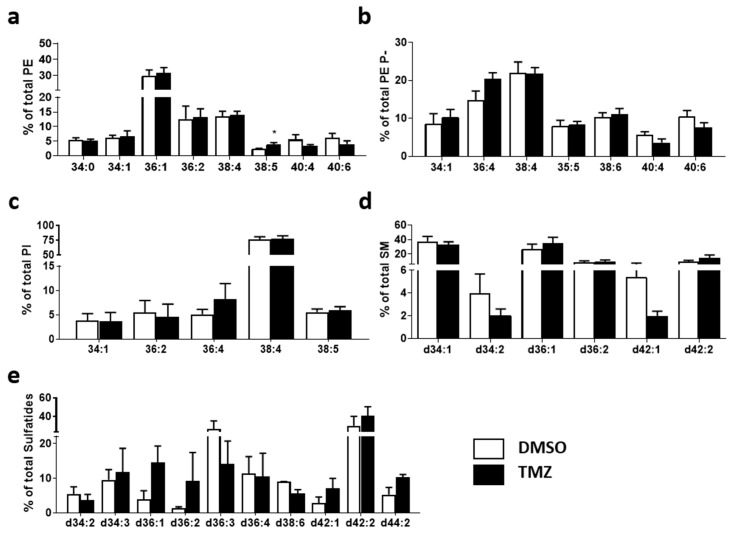
Impact of TMZ treatment on membrane lipid species of the MKI76+ region in GBM biopsies. Healthy brain and GBM biopsy sections were prepared after incubating fresh biopsies in DMSO (vehicle) or TMZ (10 mg/mL, 4 h) and analyzed by MALDI-IMS at 50 µm lateral resolution. Bar diagrams compare changes in lipid composition of (**a**) PE, (**b**) PE P-, (**c**) PI, (**d**) SM, and (**e**) sulf. Values are expressed as the percentage of total lipid species in that lipid class (mol%) and represent mean ± SEM, *n* = 5. Statistical significance was assessed using *t*-test analysis. *: *p* < 0.05. For clarity, species accounting for less than 5% were excluded from the analysis. Detailed results of all comparisons and all lipid species are included in Appendix A. Abbreviations: PE, phosphatidylethanolamine; PE P-, PE plasmalogen; PI, phosphatidylinositol; SM, sphingomyelin; sulf, sulfatide.

**Figure 6 ijms-23-02949-f006:**
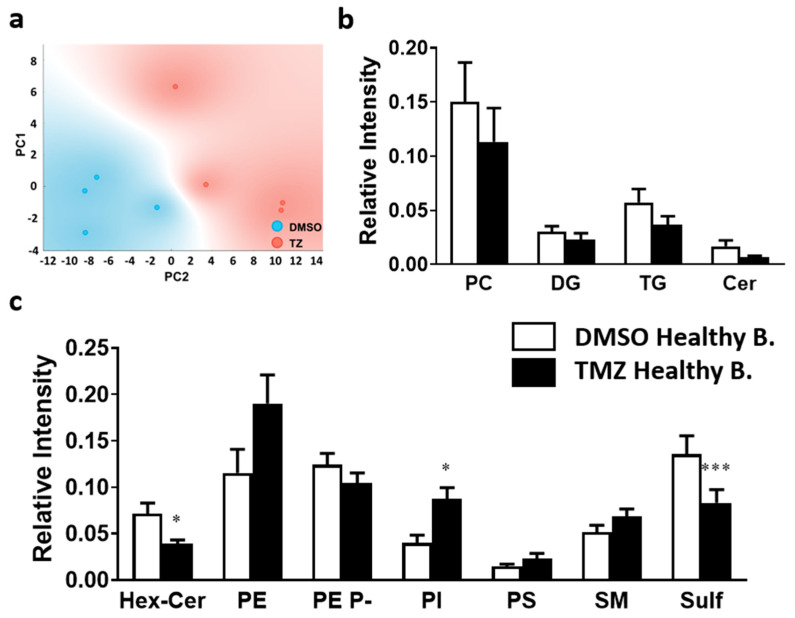
Impact of the TMZ treatment on membrane lipid classes in the MKI67+ region of healthy brain tissue. Healthy brain and GBM biopsy sections were prepared after incubating fresh biopsies in DMSO (vehicle) or TMZ (10 mg/mL, 4 h) and analyzed by MALDI-IMS at 50 µm lateral resolution. (**a**) PCA analysis of the proliferation clusters selected. There was a clear separation between control healthy brain (DMSO) (blue), and TMZ treated healthy brain (red), implying distinct lipid fingerprints. (**b**) Lipid class relative intensity was analyzed in positive-ion mode variation of selected regions in control healthy brain (DMSO) and TMZ treated healthy brain. (**c**) Lipid class relative intensity was analyzed in negative-ion mode variation of selected regions in control healthy brain (DMSO) and TMZ treated healthy brain. Values are expressed as mean ± SEM (*n* = 4). Statistical analysis was assessed using a *t*-test analysis. The asterisk (*) indicates a significant difference between both conditions * *p* < 0.05; *** *p* < 0.001. Abbreviations: HexCer, hexosylceramides; PC, phosphatidylcholine; PE, phosphatidylethanolamine; PE P-, PE plasmalogen; PI, phosphatidylinositol; PS, phosphatidylserine; SM, sphingomyelin; Sulf, sulfatide; PC, phosphatidylcholine; DG, diacylglycerol; TG, triacylglycerol; Cer, ceramide.

**Figure 7 ijms-23-02949-f007:**
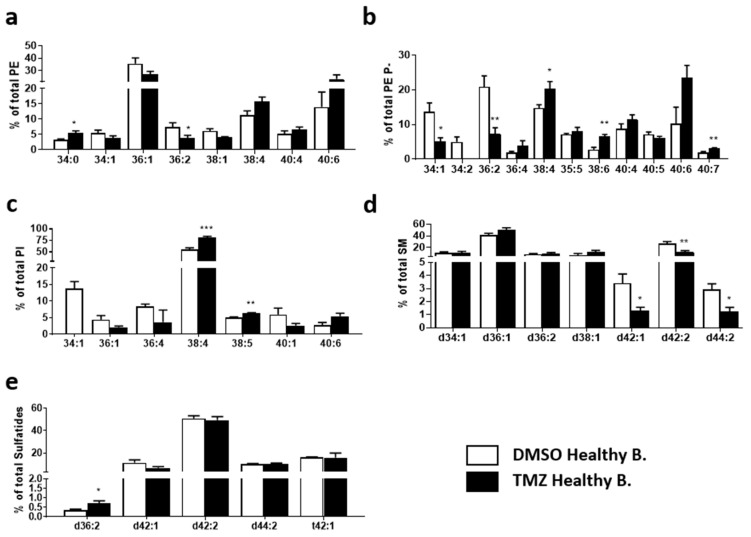
Impact of TMZ treatment on membrane lipid species in the MKI67+ region of healthy brain biopsies. Healthy brain and GBM biopsy sections were prepared after incubating fresh biopsies in DMSO (vehicle) or TMZ (10 mg/mL, 4 h) and analyzed by MALDI-IMS at 50 µm lateral resolution. Bar diagrams compare changes in lipid composition of (**a**) PE, (**b**) PE P- (**c**) PI, (**d**) SM, and (**e**) sulfatides (Sulf). Values are expressed as the percentage of total lipid species (mol%) and represent mean ± SEM, *n* = 5. Statistical significance was assessed using a *t*-test analysis. The asterisk (*) indicates a significant difference between both conditions. * *p* < 0.05; ** *p* < 0.01, *** *p* < 0.001. For clarity, species accounting for less than 5% were not included. Detailed results of all comparisons and all lipid species are included in Appendix A. Abbreviations: PE, phosphatidylethanolamine; PE P-, PE plasmalogen; PI, phosphatidylinositol; SM, sphingomyelin; Sulf, sulfatide.

**Figure 8 ijms-23-02949-f008:**
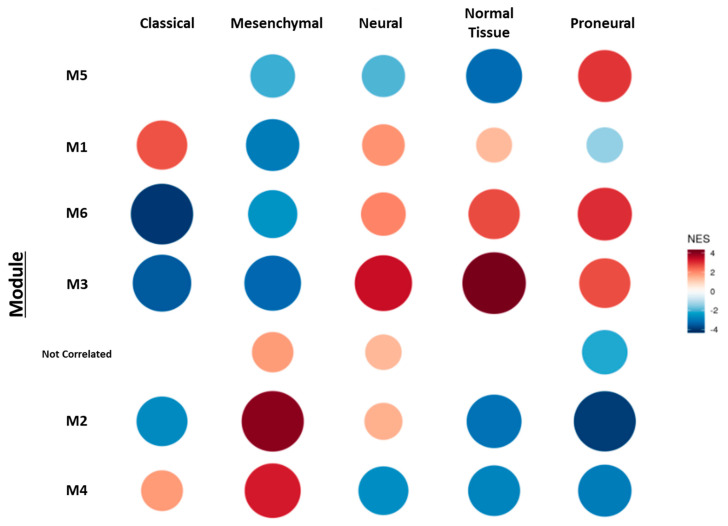
TCGA-GBM transcriptome data (AffyU133a) co-expression modular analysis of the four molecular subtypes described by Verhaak et al. [47], including normal tissue samples. Module M1 presented a high normalized enrichment score (NES) with the Classical molecular subtype, modules M2 and M4 with the Mesenchymal subtype, M5 and M6 with the Proneural subtype, and M3 with the Neural subtype and Normal samples. Module NES data and respective adjusted *p*-values (Benjamini–Hochberg) can be found in Appendix A.

**Figure 9 ijms-23-02949-f009:**
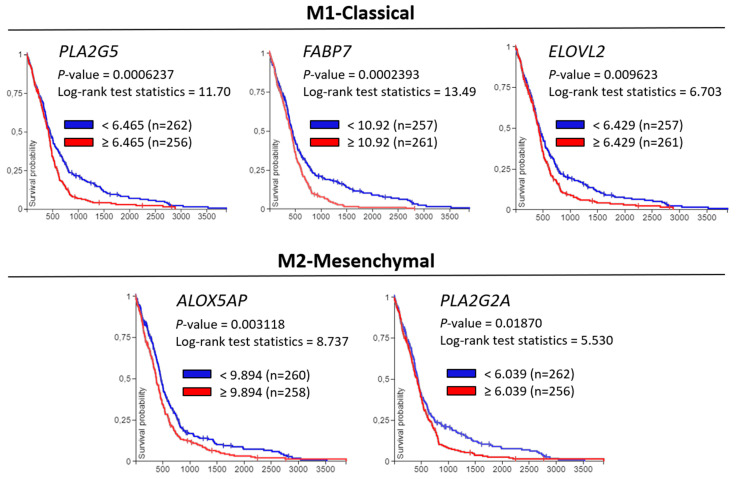
Kaplan–Meier survival analysis in GBM patients based upon PLA2G5, FABP7, ELOVL2, ALOX5P, and PLA2G2A expression in primary GBM tumors. Overall survival rates based on Xena Browser [48] two groups log-rank test for transcriptome data (AffyU133a) of primary tumor samples from TCGA-GBM database. Only the gene levels included in module M1 (Classical subtype)—PLA2G5, FABP7, ELOVL2—and M2 (Mesenchymal subtype)—PLA2G2A and ALOX5AP—which were statistically associated with the poor overall survival are represented.

**Figure 10 ijms-23-02949-f010:**
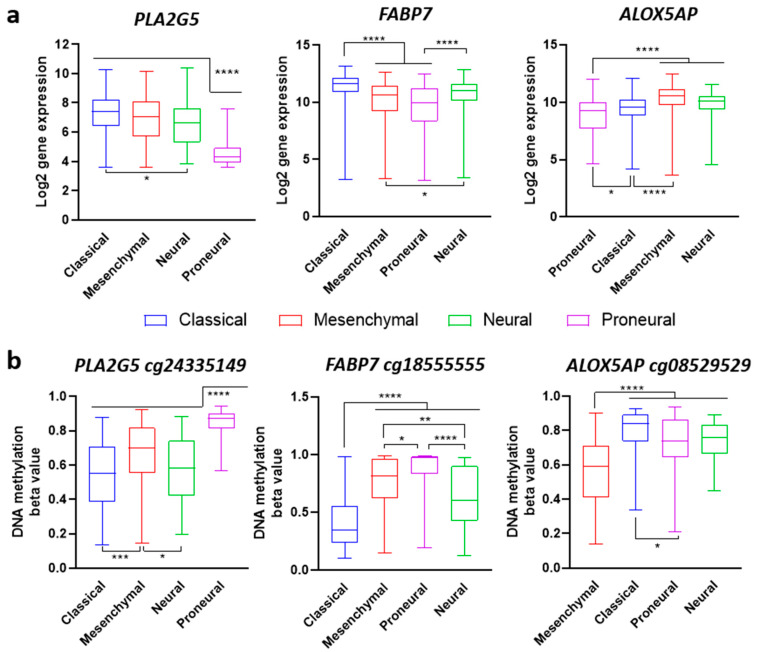
DNA methylation status and gene expression comparison of PLA2G5, FABP7, and ALOX5AP for each molecular subtype. TCGA-GBM molecular subtype dependent gene expression. (**a**) and methylation levels. (**b**) of PLA2G5, FABP7, and ALOX5AP genes. Affymetrix U133a and Methylation27k data sets from TCGA-GBM were used for data representation and statistical analysis. Statistical significance was assessed using multiple comparison ordinary one-way ANOVA with post-hoc Tukey test. * *p* < 0.05, ** *p* < 0.005, **** *p* < 0.0001.

**Table 1 ijms-23-02949-t001:** Lipid-related enzymes identified in each GBM subtype-correlated module.

		Membrane Lipid-Related Enzymes				
Classical	M1	PLA2G5	FABP7	ELOVL2		
Mesenchymal	M2	PLA2G2A	FABP5	PTGS2	ALOX15B	ALOX5AP
	M4	-				
Neural	M3	INPP5F	DGKB			
Proneural	M5	-				
	M6	UGT8				

Abbreviations: ALOX5, arachidonate 5-lipoxygenase activating protein; ALOX15B, arachidonate 15-lipoxygenase type B; DGKB, diacylglycerol kinase B; ELOVL2, fatty acid elongase 2; FABP7, fatty acid binding protein 7; FABP5, fatty acid binding protein 5; INPP5F, inositol polyphosphate-5-phosphatase F; PLA2G5, phospholipase A2 group V; PLA2G2A, phospholipase A2 groups IIA; PTGS2, prostaglandin-endoperoxide synthase 2; UGT8, UDP-galactose-ceramide galactosyltransferase.

## Data Availability

Data was obtained from TCGA Research Network and are publicly available https://www.cancer.gov/tcga.

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
