# Peer review of "Polyunsaturated Fatty Acid-Enriched Lipid Fingerprint of Glioblastoma Proliferative Regions Is Differentially Regulated According to Glioblastoma Molecular Subtype"

_ijms, 2022, doi:10.3390/ijms23062949_

Round 1
Reviewer 1 Report
The lipidome differences in GBM have attracted much attention of scientists. The authors explored how TMZ affects the lipid composition in GBM and the healthy human brain biopsies using MALDI imaging technique. The differences in lipid classes and molecular species between the GBM and healthy biopsies were compared. Interesting results of the short-term TMZ treatment experiments were obtained. The tumor lipidome was impervious to the treatment while the healthy brain was more sensitive to it. The PUFA-containing lipids in the healthy biopsies were induced multiple changes by TMZ. Furthermore, the different regulation of lipid enzymes dependent on the molecular subtype was in line with the changes in PUFA-containing phospholipids. The research is systematic and the results are significant.
Specific contents:
- Why was the TMZ treatment time just 4h? Were different treatment times tried and compared? It was unbelievable that 4h treatment causing the changes in lipids of bioposies.
- How many biopsies were detected in each experiment? And how many regions were selected in each section?
- How to identify the lipid molecular species with MALDI-IMS spectrum?
Author Response
We would like to sincerely thank the reviewers for the time taken to review our manuscript and for the constructive comments. Next, we will proceed to address, point by point, all the issues raised by the reviewers.
Response to Reviewer 1 Comments:
Observation: Moderate English changes required.
Response: Following the comment of the reviewer 1, the manuscript has been checked by a native English-speaking reviewer.
Comments and Suggestions for Authors:
The lipidome differences in GBM have attracted much attention of scientists. The authors explored how TMZ affects the lipid composition in GBM and the healthy human brain biopsies using MALDI imaging technique. The differences in lipid classes and molecular species between the GBM and healthy biopsies were compared. Interesting results of the short-term TMZ treatment experiments were obtained. The tumor lipidome was impervious to the treatment while the healthy brain was more sensitive to it. The PUFA-containing lipids in the healthy biopsies were induced multiple changes by TMZ. Furthermore, the different regulation of lipid enzymes dependent on the molecular subtype was in line with the changes in PUFA-containing phospholipids. The research is systematic and the results are significant.
Response: We appreciate very much the contextualization and comments regarding our study.
Specific comments:
Point 1: Why was the TMZ treatment time just 4h? Were different treatment times tried and compared? It was unbelievable that 4h treatment causing the changes in lipids of biopsies.
Response 1:
Our laboratory is focused on understanding the importance of lipid composition at the level of molecular species. We have previously demonstrated that commercial cell lines have a very different lipidome in comparison to the tissue they pretend to model. Furthermore, the simple fact of keeping the cells in the culture medium already alters the lipidome at the level of molecular species composition, affecting especially the distribution of PUFA containing species. For all these reasons, we decided that to carry out this study, we would use an ex vivo model, incubating human biopsies in the cell culture medium. However, there were several critical challenges to overcome.
First of all, there is a limitation associated with MALDI-technique. In our analytical setup using MALDI-IMS, tissues should be snap-frozen in liquid nitrogen in the absence of any cryoprotective substance to avoid signal interferences during the analysis. This fact, which we already know affects the integrity of harder tissues such as the colon or lung, was expected to have a similar impact, or even slightly higher impact, on a soft tissue such as the brain. In this context, it is also important to keep in mind that samples were taken in the surgical room and immediately inserted in the cell culture medium to minimize tissue degradation. There is no doubt that to keep a brain biopsy in a liquid medium at 37 ÌŠ C degrees was not optimal to maintain its integrity, and longer incubation times would have affected the structure of the tissue irreversibly. We established the time of treatment according to our experience in similar experiments using colon biopsies.
Finally, we had to deal with the scarcity of the samples. On average, this type of surgery occurs less than once-twice per month in the hospital we are working. For this reason, it was also difficult to plan to do a larger number of samples or to carry out a time-course study.
Point 2: How many biopsies were detected in each experiment? And how many regions were selected in each section?
Response 2:
We collected 5 surgical specimens of 5 different patients with GBM. We obtained a biopsy of the tumor and a biopsy of the healthy part from each specimen, which were then divided into two pieces, one to be incubated in DMSO (vehicle) and DMSO + temozolomide.
It is worth mentioning that these surgical specimens may have come from different areas of the brain, which may involve different cell distribution, particularly of the healthy area. This scenario is somehow different compared to other tissue we had analyzed previously such as a colon, where the main cell structures are rapidly identified even in malignant conditions. Finally, we took into account the importance the highly proliferative cells have in tumor progression. For all these reasons, we aimed first to identify these “highly proliferative areas” in both healthy and GBM tissue (incubated only with the vehicle) and compare the lipidomes (Fig. 1-3).
Point 3: How to identify the lipid molecular species with MALDI-IMS spectrum?
Response 3:
The lipid species were identified by comparison of their m/z with those of the lipids in the database of our software (+33,000 lipid species plus a number of possible adducts), assuming a maximum error of 0.005 Da. The group led by Dr. José Andrés Fernández, our collaborators at the University of the Basque Country have a solid experience in MALDI-IMS and in lipid identification. They have several publications where they investigate the potential pitfalls that may occur during lipid assignation. In any case, due to the intrinsic nature of the MALDI-IMS technique, where the sample is directly ionized with no previous separation, we can never rule out that more than one molecular species may contribute to the intensity of a particular peak. Hence, for those m/z for which enough signal intensity was obtained, on tissue MS/MS was carried out to confirm the initial assignment.
Reviewer 2 Report
Paper needs some revisions, look at these points:
- Lines 94-96: "Conversely, this compound induced multiple changes in the healthy brain, being the PE plasmalogen species the most affected." What do authors mean in this sentence?
- Lines 83-84: "Also, the expression of 15-HPGD (involved in prostaglandin catabolism) is associated with better outcomes in cancer patients, including GBM. " Please, discuss better this point.
- Lines 54-56: "In this context, the study of glioblastoma, as in many other cancers, is hindered due to the lack of good models to mimic the genetic heterogeneity and tumor microenvironment [5]. This is a very current topic and should be discussed more. Please improve, look at these refs: -- Decipher the Glioblastoma Microenvironment: The First Milestone for New Groundbreaking Therapeutic Strategies. Genes (Basel). 2021 Mar 20;12(3):445. doi: 10.3390/genes12030445. --- The rising stars of the glioblastoma microenvironment. Glia. 2019 May;67(5):779-79.
- Lines 71-72: "In fact, membrane lipid species are sensitive enough to be used as biomarkers for different cancer types [20–22]." Which cancers?
- Lines 89-91: "Our results showed changes in various lipid species like the phosphatidylinositols (PI)... " Are these results? if yes, they should be moved to results section.
- "we explored how TMZ affects the lipid composition in GBM and the healthy 33 human brain using imaging mass spectrometry techniques" - What is the aim of this paper? It should be expressed more clearly in the introduction section.
- There is no conclusion section. Please add.
- Lines 498-501: ".. the Proneural molecular subtype, which presents better survival data, frequently has an IDH-mutation. Moreover, IDH-mutants commonly... " It seems that recurrence pattern can influence the outcome of patients with recurrent GBM, look at these refs: -- Impact of recurrence pattern in patients undergoing a second surgery for recurrent glioblastoma. Acta Neurol Belg. 2021 Aug 16. doi: 10.1007/s13760-021-01765-4. -- Superoxide dismutase (MnSOD) in p53 expressing glioblastoma. Pathol Res Pract. 2016 Jan;212(1):17-23.
- By placing methods section after the introduction section can help readers understand your results better.
- This paper has some limitations, including the small sample size. Please state in the text.
- Lines 404-408. Is this the conclusion section ?
Author Response
We would like to sincerely thank the reviewers for the time taken to review our manuscript and for the constructive comments. Next, we will proceed to address, point by point, all the issues raised by the reviewers.
Response to Reviewer 2 Comments:
Paper needs some revisions, look at these points:
We would like to sincerely thank to the reviewer for the constructive comments
Point 1: Lines 94-96: "Conversely, this compound induced multiple changes in the healthy brain, being the PE plasmalogen species the most affected." What do authors mean in this sentence?
Response 1: The aim of this sentence was to highlight the impact of TMZ in healthy brain lipidome compared to healthy control. The alteration of healthy brain lipidome was especially notable/significant in the PE plasmalogens molecular species, being the most affected membrane lipid class.
Point 2: Lines 83-84: "Also, the expression of 15-HPGD (involved in prostaglandin catabolism) is associated with better outcomes in cancer patients, including GBM. " Please, discuss better this point.
Response 2: Prostanoids derived from the activity of cyclooxygenases and their respective synthases contribute to both active inflammation and immune response in the tumor microenvironment. Their synthesis, deactivation and role in glioma biology have not yet been fully explored and require further study. Panagopoulos AT et al (2018) (PMID: 29966699) further characterized the prostanoid pathway in grade IV glioblastoma (GBM) using quantitative real time PCR, gas chromatography/ electron impact mass spectrometry and liquid chromatography/ electrospray ionization tandem mass spectrometry. They observed significant correlations between high mRNA expression levels and poor patient survival for microsomal PGE synthase 1 (mPGES1) and prostaglandin reductase 1 (PTGR1). Conversely, high mRNA expression levels for 15-hydroxyprostaglandin dehydrogenase (15-HPGD) were correlated with better patient survival. GBMs had a higher quantity of the prostanoid precursor, arachidonic acid, versus grade II/III tumors and in GBMs a significant positive correlation was found between arachidonic acid and PGE2 content. GBMs also had higher concentrations of TXB2, PGD2, PGE2 and PGF2α versus grade II/III tumors. A significant decrease in survival was detected for high versus low PGE2, PGE2 + PGE2 deactivation products (PGEMs) and PGF2α in GBM patients. Our data show the potential importance of prostanoid metabolism in the progression towards GBM and provide evidence that higher PGE2 and PGF2α concentrations in the tumor are correlated with poorer patient survival. These findings pointed to the importance of the enzymes 15-HPGD and PTGR1 as prognostic biomarkers which could be used to predict survival outcome of patients with GBM.
We have included part of this information in the introduction (lines 94-98):
“One of these PUFAs is arachidonic acid, the precursor of a large family of bioactive molecules tightly involved in inflammation. Importantly, a study analyzing human GBM identified significant correlations between the high expression of mPGES1 and PTGR1, enzymes taking part in the synthesis of prostaglandins, and related to poor patient survival [34].
Conversely, higher gene expression of 15-HPGD, involved in prostaglandin catabolism, is associated with better outcomes in cancer patients, including GBM [34].”;
and the discussion section (lines 473-475):
“Finally, lower expression of prostaglandin synthesis enzymes, particularly PGE2S, and a lower concentration of PGE2 and PGF2α are related to better patient outcomes and lower tumor grade [34].”
Point 3: Lines 54-56: "In this context, the study of glioblastoma, as in many other cancers, is hindered due to the lack of good models to mimic the genetic heterogeneity and tumor microenvironment [5]. This is a very current topic and should be discussed more. Please improve, look at these refs: -- Decipher the Glioblastoma Microenvironment: The First Milestone for New Groundbreaking Therapeutic Strategies. Genes (Basel). 2021 Mar 20;12(3):445. doi: 10.3390/genes12030445. --- The rising stars of the glioblastoma microenvironment. Glia. 2019 May;67(5):779-79.
Response 3: We have incorporated the information included in the suggested references in the discussion (Lines 512-533). It appears as follows:
“For example, the Proneural molecular subtype, which presents better survival data, frequently has a mutation in the IDH (isocitrate dehydrogenase) gene, a well-established prognostic marker. Moreover, IDH-mutants commonly manifest the glioma CpG island methylator phenotype (G-CIMP), which is also associated with a survival advantage [74,75]. A more detailed analysis showed that in fact, enhanced survival is determined by G-CIMP level [76]. On the other hand, patients with higher MnSOD (SOD2) protein expression are most likely to be IDH1 wild type, with poor overall survival and early progression-free survival [77]. In our co-expression analysis, SOD2 was found to be one of the genes defining the M2-mesenchymal subtype (Supplementary table 4), which is characterized by poor survival, extensive necrosis, inflammation, angiogenesis, highly cell-enriched tumor micro-environment, and resistance to different therapies [78]. Further, the TCGA-GBM database shows that low SOD2 (MnSOD) expression is associated with mutant IDH1 (R132H), proneural subtype, and more specifically with G-CIMP status. Interestingly, a recent observational prospective study describes how GBM patients with wild-type IDH1/2, with Karnofsky Performance Score >80, treated with concomitant radio-chemotherapy and subsequent chemotherapy with TMZ – which presented non-local recurrence – have poorer overall survival than patients with local recurrence [79]. It might be of great interest to measure the expression of mesenchymal markers [80] in the subset of non-local recurrence patients, in order to describe a positive correlation. With this in mind, the genomic and metagenomic analysis of lipid-related enzymes could help in understanding how the observed lipid changes are regulated at the gene level.”
Point 4: Lines 71-72: "In fact, membrane lipid species are sensitive enough to be used as biomarkers for different cancer types [20–22]." Which cancers?
Response 4: As suggested by the reviewer, we have rewritten the sentence specifying the different type of cancers according to the cited studies (lines 118-120). Currently it states as follows:
“In fact, membrane lipid species are sensitive enough to be used as biomarkers for several cancer types, such as ovarian cancer, prostate cancer, and breast cancer [25–27].”
Point 5: Lines 89-91: "Our results showed changes in various lipid species like the phosphatidylinositols (PI)... " Are these results? if yes, they should be moved to results section.
Response 5: We have removed this part from the introduction as suggested by the reviewer.
Point 6: Lines 33-34: "Therefore, we explored how TMZ affects the lipid composition in GBM and the healthy human brain using imaging mass spectrometry techniques" - What is the aim of this paper? It should be expressed more clearly in the introduction section.
Response 6: We agree with the reviewer. In fact, the introduction section has been remodelled and we consider that in the current version, aims are more clearly stated.
Point 7: There is no conclusion section. Please add.
Response 7: We have added a specific conclusion section (Lines 658-679), which according to the IJMS template, should be placed after the section of Material and Methods section.
Point 8: Lines 498-501: ".. the Proneural molecular subtype, which presents better survival data, frequently has an IDH-mutation. Moreover, IDH-mutants commonly... " It seems that recurrence pattern can influence the outcome of patients with recurrent GBM, look at these refs: -- Impact of recurrence pattern in patients undergoing a second surgery for recurrent glioblastoma. Acta Neurol Belg. 2021 Aug 16. doi: 10.1007/s13760-021-01765-4. -- Superoxide dismutase (MnSOD) in p53 expressing glioblastoma. Pathol Res Pract. 2016 Jan;212(1):17-23.
Response 8: We have modified the text taking into account the references suggested by the reviewer (Lines 510-533). The text looks as follows:
The application of -omics techniques provides resourceful information to classify patients according to their molecular signature [71]. Hence, genomic, epigenomic, and transcriptomic analysis allows a precise stratification of GBM samples [72]. For example, the Proneural molecular subtype, which presents better survival data, frequently has a mutation in the IDH (isocitrate dehydrogenase) gene, a well-established prognostic marker. Moreover, IDH-mutants commonly manifest the glioma CpG island methylator phenotype (G-CIMP), which is also associated with a survival advantage [73,74]. A more detailed analysis showed that in fact, enhanced survival is determined by G-CIMP level [75]. On the other hand, patients with higher MnSOD (SOD2) protein expression are most likely to be IDH1 wild type, with poor overall survival and early progression-free survival [76]. In our co-expression analysis, SOD2 was found to be one of the genes defining the M2-mesenchymal subtype (Supplementary table 4), which is characterized by poor survival, extensive necrosis, inflammation, angiogenesis, highly cell-enriched tumor micro-environment, and resistance to different therapies [77]. Further, the TCGA-GBM database shows that low SOD2 (MnSOD) expression is associated with mutant IDH1 (R132H), proneural subtype, and more specifically with G-CIMP status. Interestingly, a recent observational prospective study describes how GBM patients with wild-type IDH1/2, with Karnofsky Performance Score >80, treated with concomitant radio-chemotherapy and subsequent chemotherapy with TMZ – which presented non-local recurrence – have poorer overall survival than patients with local recurrence [78].
Point 9: By placing the methods section after the introduction section can help readers understand your results better.
Response 9: Although we may agree with the reviewer, the manuscript structure was defined according to the MDPI-IJMS submission instructions.
Point 10: This paper has some limitations, including the small sample size. Please state in the text.
Response 10: We agree with the reviewer, and we have added some sentences regarding the limitations of the study (lines 667-672). The text appears as follows:
“However, this study has some limitations; namely, the number of samples included was limited, due to the difficulty in obtaining this type of samples and the challenges associated with the ex vivo model, which can rapidly compromise tissue structure. It is in the light of these limitations, that the results of the in-silico approach gain more relevance, as the transcriptomic and genomic results support the lipidomic results described herein.”
Point 11: Lines 404-408. Is this the conclusion section?
Response 11: Lines 404-408 should be considered as a summary the transcriptomic section, not as a conclusion of the study. Thus, lines 404-408 (currently lines 419-423) has been rewritten to make clearer its interpretation. Moreover, as mentioned in response 7, we have added a more specific conclusion section.
Round 2
Reviewer 2 Report
Authors solved all my criticisms.